# SOCS1 Deficiency Promotes Hepatocellular Carcinoma via SOCS3-Dependent CDKN1A Induction and NRF2 Activation

**DOI:** 10.3390/cancers15030905

**Published:** 2023-01-31

**Authors:** Md Gulam Musawwir Khan, Nadia Boufaied, Mehdi Yeganeh, Rajani Kandhi, Stephanie Petkiewicz, Ankur Sharma, Akihiko Yoshimura, Gerardo Ferbeyre, David P. Labbé, Sheela Ramanathan, Subburaj Ilangumaran

**Affiliations:** 1Department of Immunology and Cell Biology, Faculty of Medicine and Health Sciences, Université de Sherbrooke, Sherbrooke, QC J1H 5N4, Canada; 2Cancer Research Program, Research Institute of the McGill University Health Centre, Montréal, QC H4A 3J1, Canada; 3Department of Pathology and Laboratory Medicine, University of Ottawa, Ottawa, ON K1N 6N5, Canada; 4Harry Perkins Institute of Medical Research, QEII Medical Centre and Centre for Medical Research, Perth 6009, Australia; 5Department of Microbiology and Immunology, Keio University School of Medicine, Tokyo 160-8582, Japan; 6Department of Biochemistry, Université de Montréal, Montreal, QC H3T 1J4, Canada; 7Division of Urology, Department of Surgery, McGill University, Montréal, QC H3A 0G4, Canada; 8Centre de Recherche, Centre Hospitalier Universitaire de Sherbrooke, Sherbrooke, QC J1G 2E8, Canada

**Keywords:** hepatocellular carcinoma, SOCS1, CDKN1A, SOCS3, p53, NRF2, oxidative stress, mouse models, TCGA-LIHC

## Abstract

**Simple Summary:**

Gene coding for the SOCS1 protein is frequently inactivated in liver cancer, suggesting that SOCS1 protects liver cells from becoming cancerous. This notion is supported by the increased susceptibility of mice lacking SOCS1 to develop liver cancer. Understanding how the SOCS1 protein protects from liver cancer could help in the development of new treatment strategies. CDKN1A is another protein that protects against many cancers including liver cancer. However, under certain unknown circumstances, CDKN1A can play a completely opposite role in promoting cancer growth. When CDKN1A gains such an undesirable function is not clear. Our findings on mouse models of liver cancer, and data from human liver cancer patients, show that SOCS1 suppresses the expression of CDKN1A, and in doing so, prevents the ability of liver cells to withstand the stress associated with cancer growth. This stress reduction pathway represents a potential therapeutic target in SOCS1-less liver cancers.

**Abstract:**

SOCS1 deficiency, which increases susceptibility to hepatocellular carcinoma (HCC), promotes CDKN1A expression in the liver. High CDKN1A expression correlates with disease severity in many cancers. Here, we demonstrate a crucial pathogenic role of CDKN1A in diethyl nitrosamine (DEN)-induced HCC in SOCS1-deficient mice. Mechanistic studies on DEN-induced genotoxic response revealed that SOCS1-deficient hepatocytes upregulate SOCS3 expression, SOCS3 promotes p53 activation, and *Cdkn1a* induction that were abolished by deleting either *Socs3* or *Tp53*. Previous reports implicate CDKN1A in promoting oxidative stress response mediated by NRF2, which is required for DEN-induced hepatocarcinogenesis. We show increased induction of NRF2 and its target genes in SOCS1-deficient livers following DEN treatment that was abrogated by the deletion of either *Cdkn1a* or *Socs3*. Loss of SOCS3 in SOCS1-deficient mice reduced the growth of DEN-induced HCC without affecting tumor incidence. In the TCGA-LIHC dataset, the *SOCS1*-low/*SOCS3*-high subgroup displayed increased *CDKN1A* expression, enrichment of NRF2 transcriptional signature, faster disease progression, and poor prognosis. Overall, our findings show that SOCS1 deficiency in hepatocytes promotes compensatory SOCS3 expression, p53 activation, CDKN1A induction, and NRF2 activation, which can facilitate cellular adaptation to oxidative stress and promote neoplastic growth. Thus, the NRF2 pathway represents a potential therapeutic target in *SOCS1*-low/*SOCS3*-high HCC cases.

## 1. Introduction

The Suppressor of cytokine signaling 1 (*SOCS1*) gene is frequently repressed by CpG methylation in up to 65% of primary hepatocellular carcinoma (HCC) [1,2], suggesting that SOCS1 functions as a tumor suppressor in hepatocytes. Supporting this notion, mice lacking SOCS1 in hepatocytes show increased susceptibility to diethyl nitrosamine (DEN)-induced HCC [3,4]. *SOCS3* is also repressed in HCC albeit to a lower extent than *SOCS1* (up to 33%), and hepatocyte-specific SOCS3-deficient mice show increased DEN-induced HCC [5,6]. Even though SOCS1 and SOCS3 show close structural similarity, increased HCC development in mice lacking either SOCS1 or SOCS3 indicates their non-overlapping tumor suppressor functions [7].

SOCS1 and SOCS3 generally function as negative feedback regulators of cytokine and growth factor signaling pathways [7]. IL-6, hepatocyte growth factor (HGF), and epidermal growth factor (EGF) are critical for liver regeneration [7]. Mice lacking either *Socs1* or *Socs3* show accelerated liver regeneration, indicating that SOCS1 and SOCS3 play key roles in regulating hepatocyte proliferation [6,7,8]. Whereas SOCS3 controls IL-6 signaling, and SOCS1 regulates HGF signaling during liver regeneration [6,8,9]. As cytokines and growth factors that facilitate physiological hepatocyte proliferation also drive hepatocarcinogenesis [7], SOCS1 and SOCS3 likely mediate their tumor suppressor functions, at least partly, via attenuating HGF and IL-6 signaling, respectively.

Studies on oncogene-induced senescence implicated SOCS1 in activating p53 [10], suggesting that SOCS1 deficiency may compromise p53-mediated tumor suppression. However, SOCS1 deficiency did not diminish the induction of p53 target genes in the liver following genotoxic stress [4]. Contrarily, SOCS1 deficiency increased the expression of *Cdkn1a*, a p53 target gene, and regulated CDKN1A (p21) protein stability [4]. As a cyclin-dependent kinase (CDK) inhibitor, CDKN1A generally functions as a tumor suppressor [11]. Paradoxically, many tumors including HCC show elevated CDKN1A expression that correlates with high malignancy, poor prognosis, and drug resistance [11,12,13]. Deregulated growth factor signaling in SOCS1-deficient hepatocytes promotes AKT-mediated phosphorylation of CDKN1A, retaining it in the cytosol, where it could exert pro-tumorigenic effects [4,11]. 

Cancer growth is associated with increased cellular metabolism and oxidative stress [14]. Cancer cells escape oxidative damage by upregulating NRF2 (NFE2L2), a transcriptional activator that induces several genes including its own (*Nfe2l2*) [15,16]. The NRF2-induced proteins detoxify reactive oxygen radicals and reduce oxidative damage to macromolecules. NRF2 activity in normal cells is regulated by KEAP1, which promotes ubiquitination and proteasomal degradation of NRF2 [16]. During oxidative stress, conformational changes in KEAP1 prevent NRF2 ubiquitination, enabling newly synthesized NRF2 to escape repression by KEAP1 and induce target genes [17,18,19]. NRF2 can also be activated by proteins that interact with KEAP1 or NRF2. Notable among them are the selective autophagy substrate SQSTM1 (p62) and CDKN1A [16,20,21,22]. Here, we investigated whether CDKN1A is essential for hepatocarcinogenesis in SOCS1-deficient mice and how SOCS1 deficiency upregulates CDKN1A and promotes oncogenesis in the liver.

## 2. Materials and Methods

### 2.1. Mouse Strains

Hepatocyte-specific SOCS1-deficient mice (*Socs1^fl/fl^Alb^Cre^*) were previously described [4]. *Socs3^fl/fl^*, *Alb^Cre^*, *Cdkn1a^−/−^,* and *p53^−/−^* mice were purchased from the Jackson Laboratory. Mice lacking SOCS3, SOCS1, and SOCS3 in hepatocytes (*Socs3^fl/f^Alb^Cre^*; *Socs1^fl/fl^Socs3^fl/fl^Alb^Cre^*) were generated for this study. *Socs1^fl/fl^Alb^Cre^* mice were crossed with *Cdkn1a^−/−^* or *Tp53^−/−^* to generate SOCS1-deficient mice also lacking CDKN1A (*Socs1^fl/fl^Alb^Cre^Cdkn1a^−/−^*) or p53 (*Socs1^fl/fl^Alb^Cre^Tp53^−/−^*) in hepatocytes. All mice strains used in this study are in C57BL/6N background and are listed in Appendix A. Control mice were derived from littermates. Mice were housed in ventilated cages with 12 h day/night cycle and fed with normal chow *ad libitum*. All experiments on mice were carried out during the daytime with the approval of the Université de Sherbrooke Ethics Committee for Animal Care and Use (Protocol ID 226-17B; 2017-2043).

### 2.2. Experimental HCC

To induce HCC, DEN was administered via the intraperitoneal (i.p.) route (25 mg/kg bodyweight) into two weeks-old male mice as females are resistant to DEN-induced HCC [4,23]. All reagents and their sources are listed in Appendix A. Treated mice were sacrificed 10 months later and visible tumor nodules were counted. Tumor dimensions were measured using a digital Vernier caliper. Tumor volume was calculated in mm^3^ using the formula: (length × width^2^)/2. Liver tissues were collected in buffered formalin for histology. 

### 2.3. Induction of Genotoxic and Oxidative Stress

To induce oxidative and genotoxic stress in the liver, 6-8 weeks-old male mice were injected DEN (100 mg/kg bodyweight, i.p.) [4]. At the indicated time points, liver tissues were fixed in buffered formalin, preserved in RNAlater^®^ for gene expression analysis, or snap-frozen and stored at −80 °C to evaluate protein expression.

### 2.4. Immunofluorescence Staining of Liver Sections

To assess cell proliferation within tumor nodules, liver sections were immunostained for Ki67, and immunofluorescence (IF) images were captured by NanoZoomer and analyzed by NanoZoomer Digital Pathology (NDP) view2 software (Hamamatsu Photonics, Shizuoka, Japan). Software sources and versions are listed in Appendix A, and antibodies used for IF are in Appendix A. The number of Ki67+ nuclei were counted in 8–10 random fields for each specimen. Lipid peroxidation in DEN-treated liver tissues was evaluated by IF staining of 4-hydroxynonenol (4-HNE). Images were captured using the Axioskop 2 microscope (Carl Zeiss Canada Ltd., Toronto, Canada), and mean fluorescence intensity (MFI) was quantified (8–10 random fields/section; 3–5 mice per genotype) using the Image J software (National Institutes of Health, Bethesda, MD, USA).

### 2.5. Gene and Protein Expression Analysis

Total RNA was isolated using an RNeasy^®^ kit from liver tissues fixed in RNAlater^®^. Real-time quantitative PCR analysis was conducted as described [4] using primers listed in Appendix A. Gene expression in each DEN-treated mouse was normalized to the reference gene *Rplp0* and mRNA fold-induction was calculated relative to the expression in untreated mice of the same genotype. Preparation of cell and liver tissue lysates and western blot have been previously described [4]. Antibodies used for western blotting are listed in Appendix A.

### 2.6. TCGA-LIHC Dataset and Analyses

Transcriptomic data on The Cancer Gene Atlas-liver HCC (TCGA-LIHC) study cohort and the associated clinicopathological information [24] were downloaded (http://tcga-data.nci.nih.gov/tcga/ accessed on 3 March 2021) using a Bioconductor package TCGAbiolinks_2.14.1. [25] TCGA level 3 data comprised of 50 normal tissue, and 371 primary tumors were used after excluding three recurrent tumors.

*Gene expression analysis*: RNAseq read counts downloaded from TCGA-LIHC were normalized for sequencing depth using the size factor method implemented in a Deseq2_1.26.0 package [26]. Log2 normalized read counts were used to show gene expression levels. The significant difference in gene expression between groups was measured by the Wilcoxon test at *p* < 0.05.

*Survival analysis*: To conduct survival analysis, SOCS1 and SOCS3 expressions were converted to z-score and used to divide patients into high-expression and low-expression groups. Patients were further stratified into four groups combing SOCS1 and SOC3 expression. Kaplan-Meir survival plots were generated using the R packages survival_3.1-12 and survminer_0.4.6 [27,28]. Disease-free survival was compared between the four groups using a log-rank test. The hazard ratio was calculated via the Cox regression model using survival_3.1-12 [27].

*Pathway analysis*: “Oxygen” and “oxidant” related gene sets in the gene ontology Biological process (GO: BP) were downloaded from the Molecular Signatures Database (MsigDB) [29] using msigdbr_7.2.1 package [30]. A mod.t.test function (MKmisc 1.6 R package) [31] was used to compare each of the four patient groups (segregated based on the expression levels of SOCS1 and SOCS3) to benign samples and to score genes. Genes were then rank-ordered and gene enrichment analysis was performed using clusterProfiler_3.14.3 R package [32]. Gene sets were considered enriched with a Benjamini-Hochberg (BH) adjustment <0.05. The enrichment of the NRF2 gene, signature benchmarked by Polonen and colleagues [26], was analyzed using the Singscore 1.6.0_R package [33]. All data analysis and statistical tests were performed in R version 3.6.2 (2019-12-12).

### 2.7. Statistical Analysis

Data were analyzed using the GraphPad Prism (San Diego, CA, USA) and represented as mean ± standard error of mean (SE). Statistical significance was calculated by one-way or two-way ANOVA with Tukey’s multiple comparison test, and *p* values are represented by asterisks: * <0.05, ** <0.01, *** <0.001, **** <0.001.

## 3. Results

### 3.1. SOCS1-Mediated Tumor Suppression in the Liver Requires CDKN1A

To genetically test the potential oncogenic role of CDKN1A in SOCS1-deficient livers, we ablated *Cdkn1a* in *Socs1^fl/fl^Alb-Cre* mice and evaluated DEN-induced HCC incidence and disease severity. All *Socs1^fl/fl^Alb-Cre* mice developed numerous and large tumor nodules and showed increased liver-to-bodyweight ratio compared to *Socs1^fl/fl^* control mice (Figure 1a–e; Appendix A), confirming the tumor suppressor role of SOCS1 in hepatocytes. Mice lacking CDKN1A alone (*Socs1^fl/fl^Cdkn1a^−/−^*) also showed an increased incidence and tumor volume (Figure 1a–e; Appendix A), supporting the tumor suppressor function of CDKN1A in the liver. However, *Socs1^fl/fl^Alb-CreCdkn1a^−/−^* mice showed reduced HCC incidence with significantly fewer and smaller tumor nodules compared to *Socs1^fl/fl^Alb-Cre* or *Socs1^fl/fl^Cdkn1a^−/−^* mice (Figure 1a–e; Appendix A). These findings indicated that even though SOCS1 and CDKN1A independently function as tumor suppressors in the liver, CDKN1A promotes oncogenesis in SOCS1-deficient hepatocytes.

### 3.2. Cdkn1a Induction in SOCS1-Deficient Livers Is Driven by p53

Next, we examined the mechanisms underlying the increased expression of CDKN1A in SOCS1-deficient livers following genotoxic stress induced by DEN [4]. CDKN1A is a transcriptional target of p53 and mediates its tumor suppressor functions [11]. DEN has been reported to induce oxidative stress, DNA damage, and sustained p53 activation in the liver [34,35]. To determine whether increased Cdkn1a induction in SOCS1-deficient livers required p53, we generated Socs1^fl/fl^Alb-CreTp53^−/−^ mice and evaluated DEN-induced Cdkn1a expression. Cdkn1a was induced in Socs1^fl/fl^Alb-Cre mice several hundred-fold more strongly than in control mice and this induction was abrogated by p53 deficiency (Figure 1f). Other p53 target genes such as Mdm2, Gadd45a, Sesn1, and Sesn2 were also strongly upregulated in SOCS1-deficient livers, and this increase was also abolished by the loss of p53 (Figure 1g). These findings indicated that the increased induction of Cdkn1a in SOCS1 deficient livers is dependent on p53 activation.

### 3.3. SOCS3 Activates p53 in SOCS1-Deficient Livers

We have shown that SOCS1 promotes the activation of p53 [10,36]. SOCS3 has also been reported to activate p53 in hepatic stellate cells [37]. We have observed an upregulation of the Socs3 gene in the regenerating livers of SOCS1-deficient mice [4]. To determine whether SOCS3 could compensate for SOCS1 deficiency in promoting p53-mediated Cdkn1a induction, we first evaluated DEN-induced Socs1 and Socs3 gene expression in mice lacking either SOCS1 or SOCS3. Socs3 was upregulated nearly 16-fold in Socs1^fl/fl^Alb-Cre mice, whereas Socs1 induction in Socs3^fl/fl^Alb-Cre mice was comparable to control mice (Figure 2a). A previous study has reported increased HCC incidence in SOCS3-deficient livers [6]. However, the increased susceptibility of Socs1^fl/fl^Alb-Cre mice to HCC despite elevated Socs3 expression indicates that SOCS3 does not compensate for the loss of SOCS1 in conferring protection against HCC. Therefore, we postulated that the upregulation of SOCS3 might underlie the p53-dependent induction of Cdkn1a and promote its oncogenic activity in SOCS1-deficient livers.

To test the above hypothesis, we generated mice lacking both SOCS1 and SOCS3 in hepatocytes and examined hepatic *Cdkn1a* expression following DEN treatment. *Cdkn1a* induction in SOCS1-deficient mice was completely abrogated in *Socs1^fl/fl^Socs3^fl/fl^Alb-Cre* mice (Figure 2b). Even though *Socs3^fl/fl^Alb-Cre* mice showed significant induction of *Cdkn1a*, it occurred at a much lower magnitude than in *Socs1^fl/fl^Alb-Cre* mice (Figure 2b). SOCS3 deficiency also abrogated the induction of several other p53 targets in SOCS1-deficient livers (Appendix A). Consistent with these findings, DEN-treated *Socs1^fl/fl^Alb-Cre* mice displayed increased p53 phosphorylation and increased expression of CDKN1A (p21) protein, which were abrogated in *Socs1^fl/fl^Socs3^fl/fl^Alb-Cre* mice (Figure 2c). However, STAT3 phosphorylation, presumably resulting from IL-6 signaling, was augmented by the loss of SOCS1, SOCS3, or both (Figure 2c). These findings indicate that SOCS3 is the critical mediator of the p53-dependent *Cdkn1a* upregulation caused by genotoxic stress in SOCS1-deficient hepatocytes.

### 3.4. SOCS3 Promotes HCC Progression in SOCS1-Deficient Livers

To test whether elevated SOCS3 expression underlies the increased susceptibility of SOCS1-deficient mice to develop HCC, we evaluated DEN-induced HCC in *Socs1^fl/fl^Socs3^fl/fl^Alb-Cre* mice. Similar to *Socs1^fl/fl^Alb-Cre* mice, *Socs3^fl/fl^Alb-Cre* mice developed HCC with 100% penetrance and showed more tumor nodules than *Socs3^fl/fl^* controls (Figure 2d–h, Appendix A), confirming the non-overlapping tumor suppressor functions of SOCS1 and SOCS3 [4,6]. However, *Socs1^fl/fl^Socs3^fl/fl^Alb-Cre* mice showed significantly reduced tumor volume and liver-to-bodyweight ratio when compared to *Socs1^fl/fl^Alb-Cre* mice, even though the incidence and the number of tumor nodules were comparable between these two groups (Figure 2d–h). These data suggested that in SOCS1-deficient livers, SOCS3 does not impact tumor incidence but promotes tumor growth. In support of this notion, tumor nodules in *Socs1^fl/fl^Socs3^fl/fl^Alb-Cre* mice showed fewer Ki67-positive proliferating cells than *Socs1^fl/fl^Alb-Cre* mice (Figure 2i, Appendix A). These data indicated that SOCS3 promotes HCC progression in SOCS1-deficient hepatoma cells, possibly via the induction of p21.

### 3.5. SOCS1 Deficiency Promotes NFR2 Activation in a SOCS3 and CDKN1A Dependent Manner

Previously we have shown that SOCS1 deficiency promotes cytosolic accumulation of CDKN1A [4]. Cytosolic p21 has been implicated in activating NRF2, [20] a transcriptional activator of antioxidant response genes, and is exploited by cancer cells to counter the oxidative stress associated with neoplastic growth [16,38]. Indeed, NRF2 promotes hepatocarcinogenesis, as NRF2 deficiency in mice has been shown to confer resistance to DEN-induced HCC [39,40]. Therefore, we examined the expression of NRF2 in SOCS1-deficient mice following treatment with DEN, which induces oxidative stress in hepatocytes [41]. DEN treatment markedly increased *Nfe2l2* mRNA coding for NRF2 and its protein expression in *Socs1^fl/fl^Alb-Cre* mice which coincided with elevated p21 expression (Figure 3a,b). The increased NRF2 expression in SOCS1-deficient livers was associated with the induction of many NRF2 target genes, whereas the expression of *Keap1* was not altered (Figure 3a,c). As the upregulation of CDKN1A SOCS1-deficient livers required SOCS3, we examined whether SOCS3 promotes hepatic antioxidant response in SOCS1-deficient livers. SOCS3 deletion in *Socs1^flfl^Alb-Cre* mice abolished the DEN-induced *Nfe2l2* mRNA and NRF2 protein expression and NRF2 target gene expression in the liver (Figure 3a–c). Furthermore, HCC nodules resected from *Socs1^fl/fl^Alb-Cre* mice showed increased expression of *Cdkn1a*, *Nfe2l2,* and the NRF2 target genes *Gstm4, Gclc,* and *Nqo1*, all of which showed lower expression in HCC nodules from *Socs1^fl/fl^Socs3^fl/fl^Alb-Cre* mice (Figure 3d). These findings indicate that SOCS3 upregulates NRF2 expression and its transcriptional activity in SOCS1-deficient hepatocytes under conditions of increased oxidative stress.

Next, we examined whether SOCS3-dependent NRF2 activation in SOCS1-deficient hepatocytes required CDKN1A. DEN-induced upregulation of NRF2 mRNA and protein expression and the induction of most of the NRF2 target genes were significantly diminished in *Socs1^fl/fl^Alb-CreCdkn1a^−/−^* mice compared to *Socs1^fl/fl^Alb-Cre* mice (Figure 3e,f). Liver sections from DEN-treated wildtype mice displayed increased 4-HNE staining indicative of lipid peroxidation, which was significantly increased by SOCS1 deficiency (Figure 3g,h), reflecting the ability of SOCS1-deficient hepatocytes to withstand increased oxidative stress. CDKN1A plays a key role in this process as the increased 4-HNE staining in SOCS1-deficient hepatocytes was attenuated by simultaneous ablation of *Cdkn1a* (Figure 3g,h). These findings indicate that activation of the SOCS3-CDKN1A axis in SOCS1-deficient hepatocytes promotes NRF2-mediated antioxidant response that can increase tolerance to oxidative stress and tumor growth. 

### 3.6. SOCS1-Low/SOCS3-High HCC Cases Display Enrichment of NRF2 Signature Genes and Predict Poor Progression-Free Survival

To study the relationship between the expression levels of *SOCS1*, *SOCS3,* and *CDKN1A* in human HCC, we analyzed the TCGA-LIHC transcriptomic data [24]. The dichotomization of the TCGA-LIHC cohort based on *SOCS1* or *SOCS3* expression revealed an elevated level of *CDKN1A* expression in both *SOCS1-high* and *SOCS3-high* groups (Appendix A). As *Cdkn1a* induction in SOCS1-deficient hepatocytes required SOCS3, we stratified HCC patients based on both *SOCS1* (low, high) and *SOCS3* (low, high) expression. The *SOCS1-low*/*SOCS3-high* group represented about 15% of *SOCS1-low* HCC cases (Appendix A) and showed a significantly elevated *CDKN1A* expression compared to the *SOCS1-low/SOCS3-low* group (Figure 4a). 

Next, we performed gene set enrichment analysis (GSEA) for gene ontology (GO) terms containing ‘oxygen’ and ‘oxidant’ for each of the HCC groups compared to normal liver tissues (Figure 4b and Appendix A). The *SOCS1-low/SOCS3-low* group showed significant negative enrichment for the GO term ‘antioxidant activity’ (GO: 0016209) with a normalized enrichment score (NES) of −1.6588757 (*p* 3 × 10^−4^; *p*.*adjusted* = 7 × 10^−4^) (Figure 4c). The positive enrichment scores of *SOCS1-low/SOCS3-high* and *SOCS1-high/SOCS3-low* groups for antioxidant activity were not significant (Figure 4c, Appendix A). Curiously, the *SOCS1-high/SOCS3-high* group also showed a negative enrichment (Appendix A), although it was not significant. Notably, for the GO term ‘cellular response to increased oxygen levels’ (GO:00366296), only the *SOCS1-low/SOCS3-high* group showed a positive enrichment (NES = 1.7385050; *p* = 1 × 10^−4^; *p*.*adjusted* = 0.0011) (Figure 4d, Appendix A). Importantly, the benchmark NRF2 signature curated by Polonen and colleagues [42] revealed a highly significant enrichment in the *SOCS1-low/SOCS3-high* HCC group (NES = 1.4661141; *p* = 1 × 10^−4^; *p*.*adjusted* = 5 × 10^−4^) (Figure 4e). The *SOCS1-high/SOCS3-high* group also showed enrichment for this gene set (NES = 1.471197; *p* = 0.021; *p*.*adjusted* = 0.0104) (Appendix A). The *SOCS1-low/SOCS3-high* group showed a significantly higher enrichment of the Polonen gene signature than the *SOCS1-low/SOCS3-low* group (Figure 4f). These data indicate that increased SOCS3 expression in *SOCS1-low* human HCC is associated with elevated expression of NRF2 signature genes.

Next, we evaluated the impact of high SOCS3 expression on disease severity in *SOCS1-low* human HCC. Whereas the low *SOCS1* expression level predicted poor disease-free survival, the *SOCS3* expression level did not correlate with patient survival (Appendix A). On the other hand, high *SOCS3* expression among the low *SOCS1* expressing HCC cases predicted a shorter progression-free survival, suggesting a faster disease progression in the *SOCS1-low/SOCS3-high* patient group (Figure 4g). This notion is further strengthened by a trend towards a higher hazard ratio (HR) for this group in the univariate analysis (HR = 1.80, 95% confidence interval (CI) 0.99–3.3; *p* = 0.053; Figure 4h). These findings corroborate with the adverse impact of high *SOCS3* expression on tumor progression in *SOCS1-low* HCC, as seen in the genetically engineered *Socs1^fl/fl^Socs3^fl/fl^Alb-Cre* mouse model (Figure 2d–h). 

## 4. Discussion

In this study, we show that SOCS1 and SOCS3 function as independent tumor suppressors in the liver, however, SOCS1 deficiency in hepatocytes renders SOCS3 oncogenic in a mouse genetic model. We also provide evidence for the oncogenic role of SOCS3 in *SOCS1-low* HCC patients. Our findings show that SOCS3-mediated induction of CDKN1A contributes to NRF2 activation and cellular adaptation to increased oxidative stress associated with neoplastic growth of SOCS1-deficient HCC (Figure 5).

Clinical data and genetic models support independent tumor suppressor functions of SOCS1 and SOCS3 in the liver [1,2,3,5,6]. SOCS3 is crucial to control IL-6 signaling, whereas SOCS1 regulates HGF signaling in the liver [6,7,8,9]. SOCS1 and SOCS3 promote transcriptional activation of p53 [10,36,37], suggesting that loss of SOCS1 or SOCS3 could impact the tumor suppressor functions of p53. However, SOCS1 deficiency did not attenuate p53 activation by DEN [4]. In the present study, we show that the combined loss of SOCS1 and SOCS3 compromises the induction of p53 target genes following genotoxic stress. An additional tumor suppressor mechanism of SOCS1 could be the attenuation of the oncogenic potential of CDKN1A [4]. The current investigation, aimed to determine whether CDKN1A is essential or dispensable for HCC induction in SOCS1-deficient mice, revealed an intricate interplay between SOCS3, p53, and CDKN1A that contributes to oncogenic NRF2 activation in SOCS1-deficient hepatocytes.

CDKN1A functions as a tumor suppressor by blocking cell cycle progression via inhibiting CDK1 and CDK2, and CDKN1A-deficiency promotes spontaneous and induced tumors in different tissues [11,43,44,45]. Ablation of *Cdkn1a* in hepatocyte-specific NEMO-deficient mice, which develop chronic hepatitis, leads to spontaneous HCC [46]. We show that CDKN1A deficiency increases DEN-induced HCC (Figure 1a–e). However, elevated CDKN1A expression occurs in many cancers including HCC that correlates with poor prognosis, suggesting an oncogenic role of CDKN1A [11,47,48,49]. Indeed, CDKN1A is needed for HCC development under conditions of mild inflammation in *Fah^−/−^* mice and chronic cholestatic liver injury in *Mdr2^−/−^* mice [39,49]. These findings confirm that CDKN1A is an ‘oncojanus’ that can either inhibit or promote HCC in different contexts. CDKN1A gains oncogenic potential when retained in the cytosol via AKT-dependent phosphorylation where it interferes with apoptotic mediators [11]. CDKN1A can also promote CDK4 activity and facilitate the cell cycle progression through the G1 phase [50]. Indeed, CDKN1A promotes HCC in *Mdr2^−/−^* mice, via facilitating CyclinD-CDK4 complex formation and cell cycle progression [49]. Even though the oncojanus role of CDKN1A has been firmly established, conditions that promote its oncogenic potential remain poorly understood. Epigenetic repression of *SOCS1* in HCC can contribute to CDKN1A-mediated oncogenesis by at least two mechanisms. Increased AKT activation downstream of deregulated growth factor signaling in SOCS1-deficient livers results in CDKN1A phosphorylation and cytosolic retention [4]. As shown in the present study, a compensatory increase in SOCS3 expression can lead to increased *CDKN1A* expression via p53 activation. Even though disabling p53 mutations occur in a third of non-aflatoxin-induced HCC [51], the oncogenic pathway driven by SOCS3-p53-CDKN1A axis could still occur in a considerable proportion of HCC cases with low *SOCS1* and intact *SOCS3* expression. In support of this notion, *SOCS1-low/SOCS3-high* cases in the TCGA-LIHC cohort show poor progression-free survival. 

Cytosolic CDKN1A can bind NRF2 and relieve the inhibitory KEAP1-NRF2 interaction leading to its activation [16,20]. NRF2 is another oncojanus that enables cancer cells to cope with oxidative stress during cancer progression [16]. NRF2 is also activated by the selective autophagy substrate p62 (SQSTM1), which interrupts NRF2-KEAP1 interaction [21]. Reduced HCC development in *Fah^−/−^* mice was attributed to compensatory induction of Sestrin2, which promotes p62-dependent autophagic degradation of KEAP1, leading to NRF2 activation, protection from oxidative damage, and neoplastic transformation [39]. Contrary to the induction of NRF2 target genes in *Cdkn1a^−/−^Fah^−/−^* mice, *Cdkn1a^−/−^Socs1^−/−^* mice showed impaired induction of NRF2 target genes. While CDKN1A attenuates the antitumor functions of NRF2 in FAH-deficient livers [39], our findings indicate that CDKN1A promotes the oncogenic potential of NRF2 in SOCS1-deficient livers. These contrasting effects of CDKN1A-mediated NRF2 activation likely reflect the duality of NRF2 functions, being protective in normal and preneoplastic stages but detrimental in transformed hepatocytes [52,53].

NRF2-deficient mice have clearly established an oncogenic role for NRF2 activation in DEN-induced HCC [40]. Studies have reported that NRF2-high HCC patients show poor survival [54,55]. Even though mutations that disrupt KEAP-NRF2 interactions contribute to NRF2 activation in human and rodent HCC [56,57], NRF2 activation in HCC can also occur by other mechanisms such as elevated expression of p62 [58]. Our findings add CDKN1A induced by SOCS1 deficiency to this list of NRF2 activators in cancer cells. In addition to conferring protection from oxidative stress, NRF2 activation in hepatocytes can also provide growth stimuli via AKT activation and induction of PDGF and EGF that could promote carcinogenesis [59]. As growth factor-induced AKT activation contributes to CDKN1A phosphorylation and its cytosolic retention [4]. NRF2 can establish a positive feedback loop in CDKN1A-overexpressing cancers. SOCS1 deficiency can amplify this loop by upregulating CDKN1A expression and increasing AKT activation through increased growth factor signaling [4,9].

Overall, we have discovered an unexpected oncogenic role for SOCS3 in SOCS1-deficient HCC using mouse genetic models, for which we also found evidence in the TCGA-LIHC cohort. Targeting the pro-tumorigenic potential of SOCS3 or CDKN1A in HCC without compromising their antitumor activities will be as challenging as targeting NRF2 [52]. However, anticancer drugs activated by NRF2-induced enzymes such as NQO1 [60] could be exploited in *SOCS1-low/SOCS3-high* HCC and other instances of NRF2-mediated cancer progression.

## 5. Conclusions

SOCS1 and SOCS3 function as independent tumor suppressors in hepatocytes. However, in the absence of SOCS1, compensatory SOCS3 expression promotes p53 activation, CDKN1A induction, and NRF2 activation, facilitating cellular adaptation to oxidative stress that accompanies neoplastic growth. Thus, a key tumor suppression mechanism of SOCS1 is to prevent the tumor suppressors SOCS3 and CDKN1A from gaining oncogenic potential. The NRF2-mediated oxidative stress response induced by CDKN1A could be a potential therapeutic target in SOCS1-deficient HCC.

## Figures and Tables

**Figure 1 cancers-15-00905-f001:**
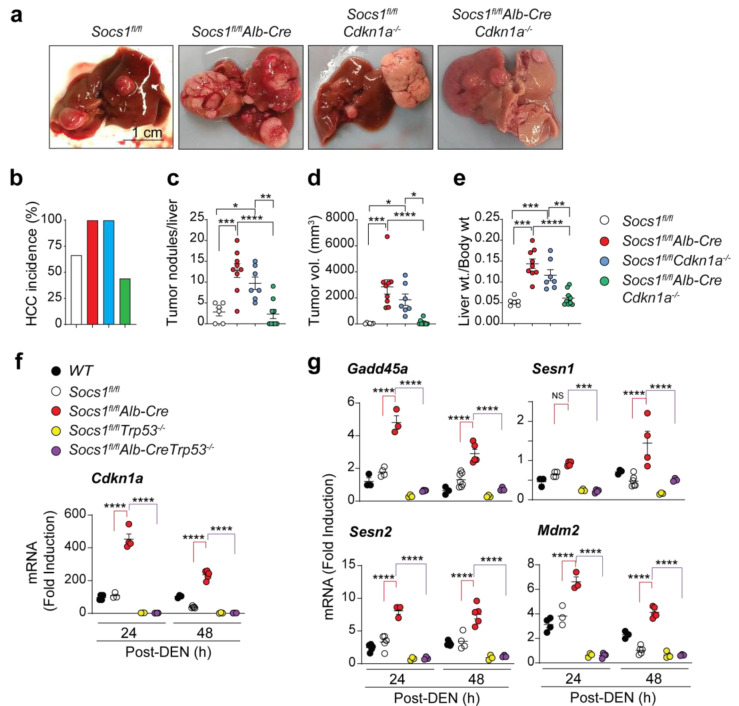
Increased HCC susceptibility of SOCS1-deficient mice requires CDKN1A induced by p53. (**a**) Two weeks-old mice of the indicated genotypes were treated with DEN (25 mg/kg bodyweight). HCC development was evaluated 10 months later. (**a**) Macroscopic liver images from representative mice for each genotype are shown (n = 5–8 mice/group). Additional data are shown in Appendix A. (**b**–**e**) Cumulative data on HCC incidence (**b**), the number of visible tumors per liver (**c**), tumor volume (sum of all tumors ≥ 2 mm in diameter; (**d**) and the liver to bodyweight ratio €. (**f**,**g**) Eight weeks-old mice (n = 3–6/group) were treated with DEN (100 mg/kg bodyweight). After 24 h and 48 h, induction of *Cdkn1a* (**f**) and the indicated p53 target genes (**g**) in the liver was evaluated by RT-qPCR. Mean ± SE; One-way (**c**–**e**) or two-way (**f**,**g**) ANOVA with Tukey’s multiple comparison tests: * *p* < 0.05, ** *p* < 0.01, *** *p* < 0.001, **** *p* < 0.0001; NS, not significant. The statistical differences are indicated to highlight the comparisons between control, SOCS1 knockout, and SOCS1p53 double knockout mice.

**Figure 2 cancers-15-00905-f002:**
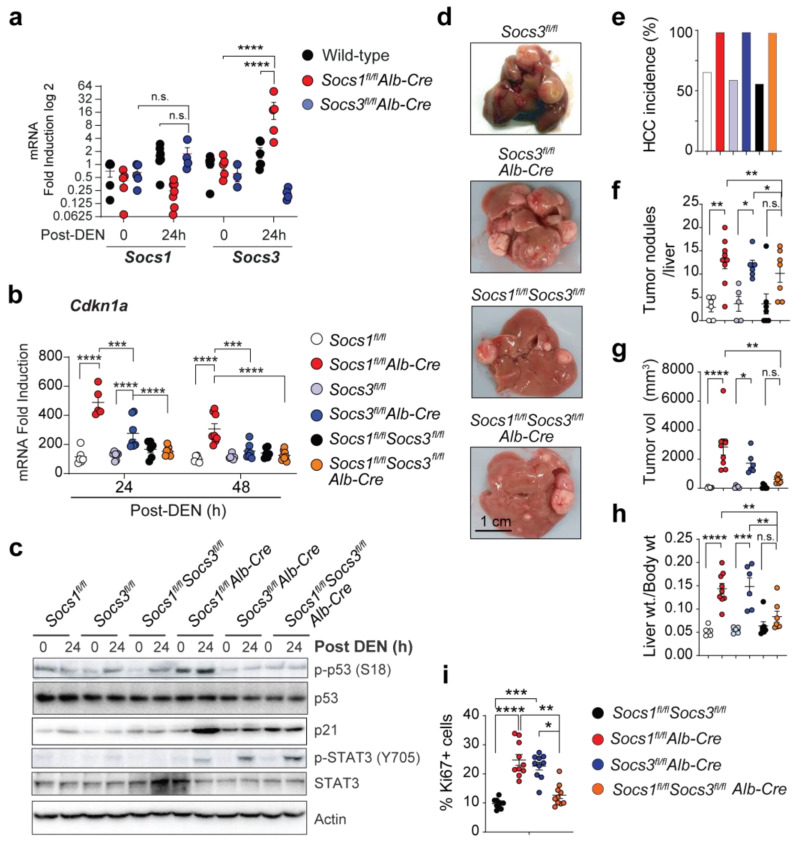
Increased HCC susceptibility of SOCS1-deficient mice requires SOCS3-dependent upregulation of CDKN1A. (**a**) Hepatic Socs1 and Socs3 gene expression in hepatocyte-specific SOCS1- or SOCS3- deficient mice (n = 5 mice/group). (**b**) RT-qPCR evaluation of hepatic Cdkn1a expression in DEN-treated mice (n = 3–8/group). (**c**) Immunoblot analysis of phospho-p53, total p53, p21, phospho-STAT3, total STAT3, and α-tubulin in control and DEN-treated mice livers (n > 3). (**d**–**h**) Two weeks-old mice of the indicated genotypes were treated with DEN (25 mg/Kg bodyweight). HCC development was evaluated 10 months later. (**d**) Representative macroscopic liver images (n = 5–9 mice/group). Cumulative data on the incidence rate of HCC (**e**), the number of visible tumor nodules (**f**), tumor volume (**g**) and the liver-to-bodyweight ratio (**h**). To facilitate comparison, HCC data for Socs1^fl/fl^ and Socs1^fl/fl^Alb-Cre mice are duplicated from (**b**–**e**). (**i**) Increased cell proliferation within SOCS1 deficient HCC tumor nodules is driven by SOCS3. Proportions of Ki67+ nuclei over Hoechst-stained nuclei (10 randomly selected areas from 3 mice/group) are shown. Mean ± SE; Two-way (**a**,**b**) or one-way (**f**–**i**) ANOVA with Tukey’s multiple comparison test; * *p* < 0.05, ** *p* < 0.01, *** *p* < 0.001, **** *p* < 0.0001; n.s. not significant.

**Figure 3 cancers-15-00905-f003:**
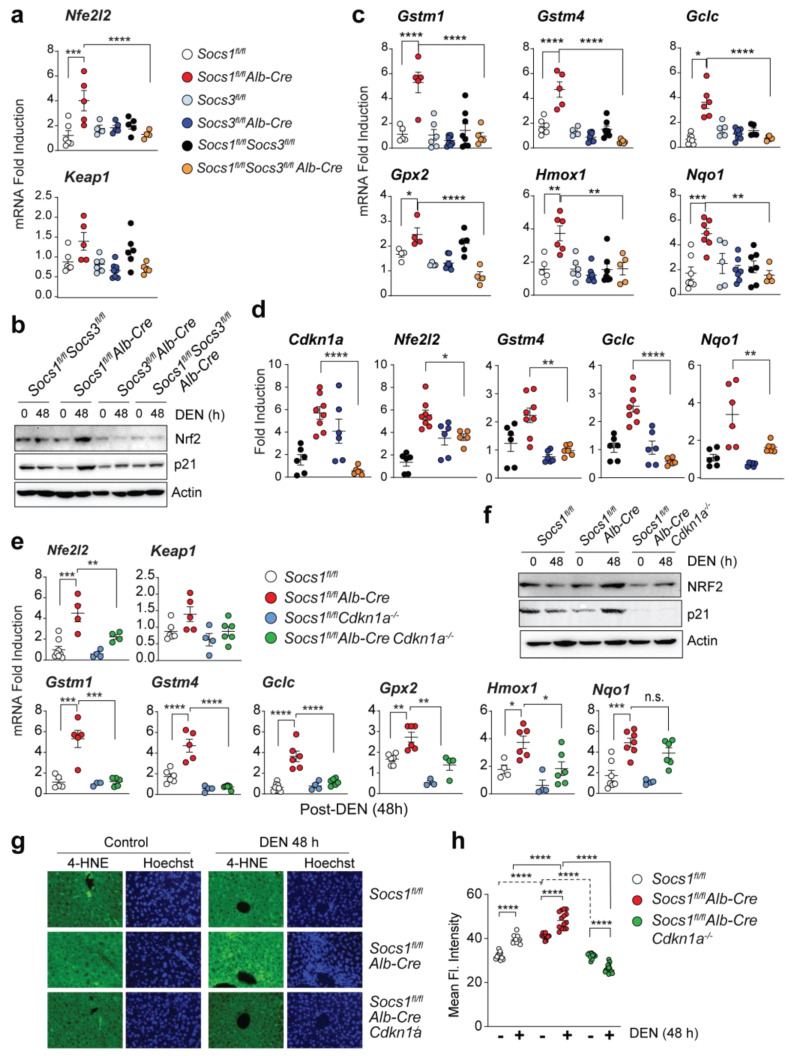
SOCS1 deficiency upregulates NRF2 and its target genes in a SOCS3 and CDKN1A-dependent manner. (**a**,**b**) Induction of the Nfe2l2 gene (**a**) and NRF2 protein (**b**) in the liver tissues of mice lacking SOCS1, SOCS3, or both in hepatocytes 48 h after DEN treatment. Cumulative data from 3–8 mice/group are shown in (**a**). For (**b**), representative data from more than two experiments are shown. (**c**) Induction of NRF2 target genes in mice lacking SOCS1, SOCS3, or both in hepatocytes 48 h after DEN treatment (n = 3–8 mice per group). (**d**) Expression of Cdkn1a, Nfe2l2, and NRF2 target genes in microscopically dissected DEN-induced HCC tumor nodules resected from the indicated genotypes of mice (n = 5–8 mice/group). (**e**) DEN-induced Nfe2l2 and NRF2 target gene expression in SOCS1-deficient mice lacking CDKN1A. Gene expression data for Socs1^fl/fl^ and Socs1^fl/fl^Alb-Cre are duplicated from (**a**,**c**) for comparison. (**f**) Immunoblot analysis of NRF2, p21, and actin in the livers of DEN-treated mice of the indicated genotypes (n > 3). (**g**) 4-HNE staining for lipid peroxidation in DEN-treated mice livers (40× magnification). Representative images from more than 3-4 mice per group are shown. (**h**) Quantification of 4-HNE staining from 3–4 mice/group. (**a**,**c**,**d**,**e**,**h**) Mean ± SE; One-way or two-way ANOVA with Tukey’s multiple comparison test; * *p* < 0.05, ** *p* < 0.01, *** *p* < 0.001, **** *p* < 0.0001.

**Figure 4 cancers-15-00905-f004:**
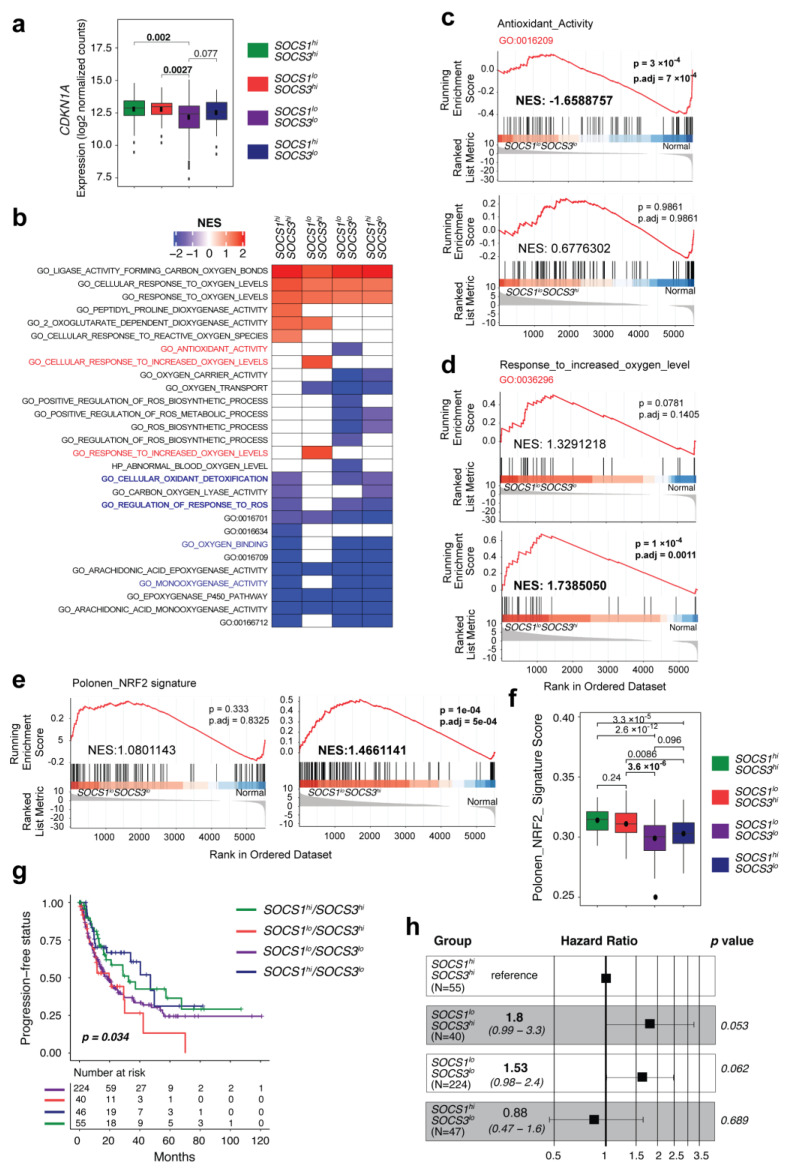
High SOCS3 expression in SOCS1-low human HCC cases is associated with elevated CDKN1A expression, enrichment of NRF2 signature genes, and fast disease progression. (**a**) Increased CDKN1A expression in SOCS1-low/SOCS3-high HCC than in SOCS1-low/SOCS3-low group. (**b**) Gene set enrichment analysis (GSEA) for biological pathways containing the term ‘oxygen’ and ‘oxidant’ in the indicated HCC groups compared to normal liver tissues. GO terms that are indicated only by numbers are given names in Appendix A. (**c**–**e**) GSEA plots showing negative enrichment of Antioxidant_Activity’ pathway genes in SOCS1-low/SOCS3-low group (**c**) and positive enrichment of ‘Response_to_increased_oxygen_level’ pathway (**d**) and ‘Polonen_NRF2 Signature’ genes (**e**) in the SOCS1-low/SOCS3-high group. (**f**) Comparison of the enrichment of Polonen_NRF2 Signature genes in the indicated HCC subgroups. (**g**) Kaplan-Meier survival plot of the TCGA-LIHC cohort grouped based on SOCS1 and SOCS3 expression levels. (**h**) Univariate analysis showing an increased hazard ratio for the SOCS1-low/SOCS3-high HCC group within the TCGA-LIHC cohort.

**Figure 5 cancers-15-00905-f005:**
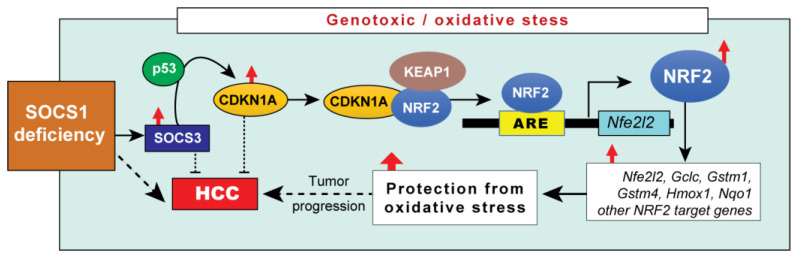
Mechanisms underlying increased HCC development in SOCS1-deficient hepatocytes. Under conditions of genotoxic and oxidative stress, SOCS1-deficient hepatocytes upregulate SOCS3, presumably to compensate for the loss of certain SOCS1-dependent functions. However, SOCS3 promotes p53 activation and induces CDKN1A, which interacts with NRF2, leading to NRF2 activation and induction of NRF2 target genes. As protein products of these genes confer protection from oxidative stress, they could facilitate the adaptation of SOCS1-deficient HCC to increased cancer cell metabolism and disease progression.

## Data Availability

The materials and the data that support the findings of this study are available from the authors upon reasonable request.

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
