# Peer review of "SOCS1 Deficiency Promotes Hepatocellular Carcinoma via SOCS3-Dependent CDKN1A Induction and NRF2 Activation"

_cancers, 2023, doi:10.3390/cancers15030905_

Round 1

Reviewer 1 Report

In the present study, the authors investigated the potential mechanism of SOCS3-mediated HCC exacerbation under the SOCS1 deficient condition. In vivo analyses using the DEN-induced HCC mouse model demonstrated that SOCS3 could contribute to HCC development via the p53-Cdkn1a signaling. It was also demonstrated that Cdkn1a was involved in the upregulation of NRF2/Nfe2l2 and NRF2-induced antioxidant response genes that generally increase tolerance to oxidative stress and tumor growth. In vitro analyses using Hapa1-6 and Hep3B cell lines showed consistent results. Further, a data mining analysis of the TCGA-LIHC dataset showed that the CDKN1A expression level was high in the SOCS1-low/SOCS3-high subgroup, associated with enrichment of the NRF2 transcriptional signature, faster disease progression, and poor prognosis. According to these results, the authors suggested that the NRF2 pathway represents a potential therapeutic target in SOCS1-low/SOCS3-high HCC cases.

These results would provide novel insights into the molecular mechanism of HCC development, diagnosis/classification, and therapeutic targets. However, there are several concerns that need to be addressed.

Major Concerns:

(1)

TCGA-LIHC data analyses showed gene alterations in tumors. On the other hand, DEN-induced in vivo gene expression changes were investigated only for the acute phase, e.g., at 24 or 48 hr post-administration. Hence, it is unclear if the gene expression changes observed at 24 or 48 hr remained in the mouse liver tumors. For instance, the Cdkn1a expression level in the Socs1-fl/fl:Alb-Cre liver appears lower at 48 hr than 24 hr in Fig 2b.  

The expression levels of Cdkn1a, NRF2/Nfe2l2, and Socs3 in the tumors developed in wild-type, Socs1-fl/fl:Alb-Cre, and Socs1-fl/fl:Socs3-fl/fl:Alb-Cre mice may need to be presented.

In addition, the reason was not clear why 8-week-old mice were used for gene expression analyses, while 2 weeks old mice were used for tumor formation assay. Could DEN induce HCC regardless of age of administration? The mechanism/pathway induced by DEN may differ between 2-week-old and 8-week-old mice. Please explain the reason for using 8-week-old mice for gene expression analyses.

(2)

In vitro analyses seem to be insufficient to support the authors’ hypothesis. 

(2a) In relation to Fig 3a-c, the expression levels of SOCS3 in Hepa-SOCS1 and Hep3B-SOCS1 need to be presented.

(2b) In relation to Fig 3a-c, the status of p53(S18)-phosphorylation in Hepa-SOCS1 and Hep3B-SOCS1 needs to be presented.

(2c) Similar to Fig 3a-c, the effect of the SOCS3 overexpression on the expression of p21/Cdkn1a and NFR2/Nfe2l2 needs to be investigated.

(2d) In relation to Fig 3j, the effect of the p21-knockdown on the expression levels of NFR2/Nfe2l2 and its target genes need to be investigated.

(3)

While DEN was used in in vivo analyses, t-BHP was used in in vitro analyses. Please explain the reason why DEN was not used in in vitro analyses. There is a possibility that DEN and t-BHP may have different effects.

Minor concerns:

(4)

Lines 132-134, ‘Stable lines of Hepa and Hep3B cells expressing SOCS1 (Hepa-SOCS1) or the control vector were previously described [8,9].’;

- Methods for the generation of Hepa-SOCS1 and Hep3B-SOCS1 were not described in 8 and 9. Please briefly describe the methods of preparation of these cell lines. 

(5)

Lines 169-170, ‘Significant difference in gene expression between groups was measured by Wilcoxon test at p <0.05.’;

- Please add the list of differentially expressed genes as supplementary data.

(6)

Fig 1c-e,

Does Fig 1d represent the total tumor volume of each liver? Were there any differences in the distribution of lesion size among the groups? 

(7)

Fig 1f-g,

- It is unclear which plot belongs to the 24 h or 48 h group.

- Only two comparisons show a significant difference. Did the other comparisons show differences?

(8)

Fig1, figure legend, ‘n.s., not significant’;

- n.s. is not indicated in the figure.

(9)

Fig 2e and 2i;

- Cdkn1a was not upregulated in the Socs1-fl/fl:Socs3-fl/fl:Alb-Cre liver at 24-48 hr post-DEN injection. The Ki67-positive cell ratio in the Socs1-fl/fl:Socs3-fl/fl:Alb-Cre liver tumor was similar to that of Socs1-fl/fl:Socs3-fl/fl. However, the HCC incidence in the Socs1-fl/fl:Socs3-fl/fl:Alb-Cre liver was 100%, higher than Socs1-fl/fl:Socs3-fl/fl liver. Please explain how to interpret these results.

(10)

Fig 3a;

- Please label what the X-axis indicates. In addition, please indicate the administrated t-BHP concentration.

(11)

Fig 3, lines 295-297, ‘(Hep3B-SOCS1) of the control vector (Hep3B-vector) were treated with tBHP for 24h, and NRF2 and p21 protein levels were evaluated at the indicated time points.’;

- The experimental condition of in vitro assays is not clear. What does timepoint 0 indicate in panels 3a, and 3e-j?

(12)

Fig 3h and 3i;

It would be better to present it in a consistent graphical style.

(13)

Fig 3j, western-blot;

- Please indicate the timepoint of analysis.

(14)

Lines 339-341 and Fig 3g;

Immunofluorescent results do not seem to show quantitative differences. Is RT-qPCR or Western-blot analyses possible?

(15)

Lines 385-389 and Fig 4g;

Similar to 14, is RT-qPCR or Western-blot analyses possible?

(16)

Line 370, ‘NRF2 mRNA’;

- Please indicate that NRF2 is a protein encoded by the Nfe2l2 gene.

(17)

Fig 5g;

- Please label what the X-axis indicates.

- Please add an explanation to the figure legend as to what the bottom table shows.

Author Response

We thank the  Reviewer for valuable comments and constructive critiques. We have carefully addressed the concerns and provide a point-by-point response below.

Major Concerns:

(1)(a) TCGA-LIHC data analyses showed gene alterations in tumors. On the other hand, DEN-induced in vivo gene expression changes were investigated only for the acute phase, e.g., at 24 or 48 hr post-administration. Hence, it is unclear if the gene expression changes observed at 24 or 48 hr remained in the mouse liver tumors. For instance, the Cdkn1a expression level in the Socs1-fl/fl:Alb-Cre liver appears lower at 48 hr than 24 hr in Fig 2b.

(b) The expression levels of Cdkn1a, NRF2/Nfe2l2, and Socs3 in the tumors developed in wild-type, Socs1-fl/fl:Alb-Cre, and Socs1-fl/fl:Socs3-fl/fl:Alb-Cre mice may need to be presented.

 (c) In addition, the reason was not clear why 8-week-old mice were used for gene expression analyses, while 2 weeks old mice were used for tumor formation assay. Could DEN induce HCC regardless of age of administration? The mechanism/pathway induced by DEN may differ between 2-week-old and 8-week-old mice. Please explain the reason for using 8-week-old mice for gene expression analyses.

 Response: We divided the abovementioned critiques into three subsections (a, b, c) to facilitate the response to (c) first, as this would make it easier to respond to (a) and (b).

(c) Biotransformation of DEN in hepatocytes generates reactive intermediates that cause methylguanine DNA adducts, which induce genotoxic response and mutations (DOI: 10.1177/0023677215570086). While this activity should not be different between neonates and adults, it is known for over 3 decades that administration of a single dose of DEN to 2 weeks-old mice consistently induces HCC. It is believed that liver development continues during the neonatal period until adulthood, and DEN exerts a ‘complete’ carcinogenic effect on proliferating hepatocytes of neonatal mice but not in quiescent hepatocytes of adult mice. Repetitive DEN administration in adults can induce HCC, but with wide variability, and thus DEN is an ‘incomplete carcinogen’ in quiescent hepatocytes. For these reasons, 2-weeks old mice are used for inducing HCC, and adult mice to study genotoxic response. The references on these observations are very old, but this information are succinctly described in the introduction section (5th paragraph) of  this article: https://doi.org/10.1016/j.cell.2005.04.014. Therefore, we used 2 weeks old mice to induce HCC by DEN and adult mice to study DEN-induced genotoxic response.

(a) Endogenously generated metabolites such as reactive oxygen and nitrogen intermediates during acute and chronic inflammatory conditions of the gut and liver induce DNA damage response. Besides, ongoing inflammation in cirrhotic livers and tumor nodules, also cause change in hepatic gene expression. The latter is captured in the TCGA dataset, which averages the gene expression in several hundred specimens. The variations seen between samples within each group could result not only from differing stress conditions in individual tumors but also from their non-uniform distribution within different parts of the resected tumor used for RNAseq.

DEN is used in mouse models to induce genotoxic stress ‘at a small time-window’ to mimic the toxic conditions and agents that can cause DNA damage in hepatocytes in humans. DEN is metabolized in hepatocytes to release toxic/carcinogenic intermediates that are eventually detoxified and eliminated. While DEN-induced mutations (which can vary between different DEN—induced tumors) that cause HCC will remain, the DEN-induced general genotoxic response will vane once DEN is metabolized and eliminated. However, in established DEN-induced tumors, tumor-induced inflammatory conditions – including ROS, RNS – can cause genotoxic stress response. (please see below – response to 1b).

The data comparing DEN-induced Cdkn1a expression (as part of DNA damage response) simply highlights the fact that this response is amplified in hepatocytes lacking SOCS1.

(b) We have provided data on the expression levels of Cdkn1a, Nfe2l2, Gstm, Gclc and Nqo1 genes in the tumors resected from wild-type, Socs1-fl/fl:Alb-Cre, and Socs1-fl/fl:Socs3-fl/fl:Alb-Cre mice in Figure 4d (now Figure 3d in the revised manuscript- please see response to the next critique #2 for this change in figure number). These data show elevated expression of these genes in HCC resected from Socs1-fl/fl:Alb-Cre  mice compared to HCC nodules from wildtype mice, and this increase was significantly reduced in tumors from Socs1-fl/fl:Socs3-fl/fl:Alb-Cre mice. This is indicated in lines 529-531 of the revised manuscript.

We have already shown that Socs3 expression in SOCS1-deficient hepatocytes is indispensable to upregulate Cdkn1a (Figure 2b) and Nfe2l2 (Figure 4a). It will be superfluous to show the expression levels of Socs3 in these tumors.

(2) In vitro analyses seem to be insufficient to support the authors’ hypothesis. 

(2a) In relation to Fig 3a-c, the expression levels of SOCS3 in Hepa-SOCS1 and Hep3B-SOCS1 need to be presented.

(2b) In relation to Fig 3a-c, the status of p53(S18)-phosphorylation in Hepa-SOCS1 and Hep3B-SOCS1 needs to be presented.

(2c) Similar to Fig 3a-c, the effect of the SOCS3 overexpression on the expression of p21/Cdkn1a and NFR2/Nfe2l2 needs to be investigated.

(2d) In relation to Fig 3j, the effect of the p21-knockdown on the expression levels of NFR2/Nfe2l2 and its target genes need to be investigated.

 Response:

We agree that all the above concerns are appropriate, and the data provided in these in vitro experiments are incomplete ‘with respect to the in vivo data’ on knockout mouse models.

Please note that some of these concerns are also raised by Reviewer #2, especially the statement that ‘We cannot rule out that SOCS1 overexpression and loss may have very different effects’, as cancer cell lines accumulate many mutations after decades of in vitro cultures that could impact p53 functions.

We have previously show that SOCS1 expression downmodulates p21 (Ref #4: DOI: 10.1038/onc.2015.485).  It has been shown that p21 promotes NRF2 activation (Ref #20: DOI: 10.1016/j.molcel.2009.04.029). Therefore, we used the cell line models solely to show that ‘SOCS1-mediated p21 downmodulation reduces NRF2 activation’. We now realize that the Revierwer-1’s major concern #2 on the cell lines data are legitimate in the light of our in vivo findings.

The requested experiments will need several months to complete, and the outcome will be uncertain because the in vitro experiments may not faithfully mirror the in vivo conditions, which is echoed by the 2nd reviewer in his/her/their comments: ‘SOCS1 overexpression and loss may have very different effects’. In addition, genetic alterations accumulated in these long-established cell lines may influence the impact of SOCS1 overexpression on Socs3 gene induction and p53 activation. In fact, Hep3B does not express p53 (Ref: DOI: 10.1007/s10616-014-9761-9).

We believe that the already published work showing the p21-driven Nrf2 activation and our in vivo data with genetic proofs that clearly demonstrate the ‘SOCS1 loss-p21 induction-NRF2 activation’ pathway (via Socs3 upregulation) are strong enough ‘to proceed from Figure 2 to Figure 4’ without the need for Figure 3. We feel that removing the in vitro cell line data will eliminate the concerns raised and make the manuscript straight forward without compromising the conclusion of this manuscript in any way. Therefore, we have eliminated Figure 3 from the revised manuscript and modified the manuscript accordingly to make a smooth transition from Figure 2 to Figure 4 (Figure 3 in the revised MS) in the Results section. Appropriate modifications are made in the abstract to reflect these changes.

(3) While DEN was used in in vivo analyses, t-BHP was used in in vitro analyses. Please explain the reason why DEN was not used in in vitro analyses. There is a possibility that DEN and t-BHP may have different effects.

 Response:

We agree that DEN (generates methylguanine DNA adducts) and t-BHP (induces oxidative stress) can exert different effects on hepatocytes.  As mentioned above, the in vitro cell line experiments were solely used to show SOCS1 regulation of p21 and Nrf2 and its impact on oxidative stress. t-BHP is the most potent oxidizing agent used to mimic oxidative stress in cells in vitro, whereas DEN is an inducer of genotoxic and oxidative stress in vivo. Even though the use of t-BHP in vivo and DEN in vitro are extremely hard to find in the literature, they can be tested. However, this question now becomes irrelevant as the cell line data are omitted in the revised manuscript.

Minor concerns:

(4) Lines 132-134, ‘Stable lines of Hepa and Hep3B cells expressing SOCS1 (Hepa-SOCS1) or the control vector were previously described [8,9].’;

- Methods for the generation of Hepa-SOCS1 and Hep3B-SOCS1 were not described in 8 and 9. Please briefly describe the methods of preparation of these cell lines. 

Response: All cell line data are removed in the revised manuscript.

(5) Lines 169-170, ‘Significant difference in gene expression between groups was measured by Wilcoxon test at p <0.05.’;

- Please add the list of differentially expressed genes as supplementary data.

Response: The statistical test mention in lines 169 and 170 refers to the boxplot (Figure 4A) where we reported the expression of CDKN1A. We used Wilcoxon test to compare log normalized read count since RNA-seq data are not normally distributed.  We did not run DEG analysis (DESeq or EdgeR methods) to identify differentially expressed genes between the different groups and gene enrichment analyses were run on the whole transcriptome, as reported in the Materials and Methods, using publicly available gene signatures.

(6) Fig 1c-e, Does Fig 1d represent the total tumor volume of each liver? Were there any differences in the distribution of lesion size among the groups? 

Response: Fig 1d shows the tumor volume, which is the sum of all tumors ≥2 mm in diameter. This is indicated in figure legends. We did not make a recording of the size distribution.

(7) Fig 1f-g, - It is unclear which plot belongs to the 24 h or 48 h group.

- Only two comparisons show a significant difference. Did the other comparisons show differences?

Response: The two sets are now indicated by separating lines at the bottom to make the distinction between 24h and 48h groups clear.

The comparison highlights the difference between SOCS1 KO and SOCS1p53 double KO. We have now added the control mice. Other comparisons are not indicated simply to reduce clutter.

(8) Fig1, figure legend, ‘n.s., not significant’;

- n.s. is not indicated in the figure.

Response: This is an omission on our part while finalizing the figures. It is now removed.

(9) Fig 2e and 2i;

- Cdkn1a was not upregulated in the Socs1-fl/fl:Socs3-fl/fl:Alb-Cre liver at 24-48 hr post-DEN injection. The Ki67-positive cell ratio in the Socs1-fl/fl:Socs3-fl/fl:Alb-Cre liver tumor was similar to that of Socs1-fl/fl:Socs3-fl/fl. However, the HCC incidence in the Socs1-fl/fl:Socs3-fl/fl:Alb-Cre liver was 100%, higher than Socs1-fl/fl:Socs3-fl/fl liver. Please explain how to interpret these results.

Response:

We recognize this issue and have already discussed it in section 3.4. The data shown in Figure 2d-2i show that in SOCS1-deficient livers, SOCS3 does not impact tumor incidence but promotes to tumor growth. This statement is now included to make it clear that SOCS3 facilitates tumor progression in SOCS1-deficient HCC (lines 353-354).

(10) Fig 3a;

- Please label what the X-axis indicates. In addition, please indicate the administrated t-BHP concentration.

Response: We regret this omission (t-BHP (150 uM, h)). However, Figure 3 on cell lines is completely removed in the revised manuscript.

(11) Fig 3, lines 295-297, ‘(Hep3B-SOCS1) of the control vector (Hep3B-vector) were treated with tBHP for 24h, and NRF2 and p21 protein levels were evaluated at the indicated time points.’;

- The experimental condition of in vitro assays is not clear. What does timepoint 0 indicate in panels 3a, and 3e-j?

Response: In Figure 3a, 3d and 3e the ‘0’ should have been NT (non-treated). Again, Figure 3 on cell lines is removed in the revised manuscript.

(12) Fig 3h and 3i;

It would be better to present it in a consistent graphical style.

Response: We agree. However, these figures are now removed in the revised manuscript.

(13) Fig 3j, western-blot;

- Please indicate the timepoint of analysis.

Response: Another regrettable omission (24 h), but the figure is now removed.

(14) Lines 339-341 and Fig 3g;

Immunofluorescent results do not seem to show quantitative differences. Is RT-qPCR or Western-blot analyses possible?

Response: HNE reactivity by western blot is not clean enough for quantification. Western blotting for 8-hydroxy-2-deoxyguanosine (8-oxo-dG) for oxidation could be used, but we have not optimized it yet. This figure is now removed.

(15) Lines 385-389 and Fig 4g;

Similar to 14, is RT-qPCR or Western-blot analyses possible?

Response: In our hands, HNE IF staining provided the most consistent results. A search in the published literature to quantify oxidative stress in FFPE tissue sections did not bring out any impressive results, even though many reagents are shown to provide convincing results in cultured cells.  We have done NRF2 IF staining on FFPE sections with reasonably good results, but the Ab batch that we used is no longer available and thus could not be reproduced.  A few other NRF2 Ab that we tested did not give satisfactory results. Therefore, we did not show the NRF2 IF data. We had similar issues with GCLC and HO-1 antibodies. At present, we do not have a better indicator of oxidative stress in FFPE tissues and are trying to find a good marker. A potential candidate is 8-hydroxy-2-deoxyguanosine (8-oxo-dG), which we need to optimize for use on liver FFPE sections.

(16) Line 370, ‘NRF2 mRNA’;

- Please indicate that NRF2 is a protein encoded by the Nfe2l2 gene.

Response: Thank you, it is now corrected.

(17) Fig 5g;

- Please label what the X-axis indicates.

- Please add an explanation to the figure legend as to what the bottom table shows.

Response: Another regrettable omission (months), which is now corrected.

Reviewer 2 Report

The authors examined the role of SOCS1 in liver cancer with a series of mouse models. Although SOCS1, SOCS3 and CDKN1A are all putative tumor suppressor genes on their own, the authors show that SOCS1 deficient livers are more susceptible to tumorigenesis through the effects of SOCS3 and CDKN1A. SOCS3 is upregulated in mouse livers lacking SOCS1 and in turn upregulates CDKN1A through p53. CDKN1A activates NRF2 and its downstream antioxidant program to protect cancer cells from oxidative and genotoxic stress. Though the study is very interesting in terms of the in vivo findings and the interplay between SOCS1 and SOCS3, some results should be clarified before the manuscript is accepted for publication.

Major concerns:

1)      In Figures 4g-h, the authors saw an increase in lipid peroxidation in Socs1fl/flAlb-Cre animals and claim that this is indicative of better tolerance for oxidative damage, though increased staining like this is often indicative of greater oxidative damage. Though a case may be made for increased tolerance, the authors’ own results in Figures 3f-g are contradictory with this claim. In Figure 3, increased expression of SOCS1 led to increased ROS and increased 4-HNE staining, which is the opposite of Figure 4. Is this a problem with the 4-HNE staining? The authors should try another dye for ROS in the liver slices of Figure 4g. If they still show increased ROS in Socs1fl/flAlb-Cre animals, the authors should explain this discrepancy.

2)      The in vitro overexpression experiments of Figure 3 do show that SOCS1 overexpression may suppress CDKN1A, but these results are also slightly puzzling and require more clarification. The authors’ previous work showed that SOCS1 promoted p53 activation (reference 10). It is strange that the p53 target CDKN1A has lower expression when SOCS1 is overexpressed in the liver cell lines. Is this CDKN1A effect dependent on p53 or is this a result of SOCS3 being downregulated when SOCS1 is overexpressed? The authors should show the expression levels of SOCS3 when SOCS1 is overexpressed. Additionally, all the in vivo experiments were done with loss of SOCS1, but the mechanistic in vitro experiments were done with SOCS1 overexpression. We cannot rule out that SOCS1 overexpression and loss may have very different effects. The authors should try to knock out or knock down SOCS1 in the liver cell lines to see if they get a similar downstream effect of CDKN1A and NRF2 activation.

Minor concern:

1)      Figure legend for Figure 4f is missing.

Author Response

We thank the Reviewer for valuable comments. We have carefully addressed the concerns and provide a point-by-point response below.

Major concerns:

1) (a) In Figures 4g-h, the authors saw an increase in lipid peroxidation in Socs1fl/flAlb-Cre animals and claim that this is indicative of better tolerance for oxidative damage, though increased staining like this is often indicative of greater oxidative damage. Though a case may be made for increased tolerance, the authors’ own results in Figures 3f-g are contradictory with this claim.

(b)In Figure 3, increased expression of SOCS1 led to increased ROS and increased 4-HNE staining, which is the opposite of Figure 4.

(c) Is this a problem with the 4-HNE staining? The authors should try another dye for ROS in the liver slices of Figure 4g. If they still show increased ROS in Socs1fl/flAlb-Cre animals, the authors should explain this discrepancy.

Response: The above concern is divided into three subsections (a, b, c) to provide pointed response to each of these questions.

(a) To our knowledge, 4-HNE is a both a product and a mediator of oxidative damage to cellular lipids (DOI: 10.1016/s0163-7827(03)00014-6; DOI: 10.3390/biom5042247). Hence, it’s increased staining intensity represents both increased oxidative stress and, in the context of Figure 4g (Figure 3g in the revised manuscript), increased tolerance to this stress. This increased tolerance and the consequential augmentation of oxidative stress are mitigated by p21-dependent NRF2-mediated antioxidant response, as the increased 4-HNE staining in SOCS1 KO liver is diminished by Cdkn1a deletion.

(b) Figure 3f and 3g are not contradictory to the data shown in Figure 4g. In Figure 3 SOCS1 overexpression decreases (not increase) 4-HNE and ROS levels. Nonetheless, in response to the concerns of both Reviewer 1 about the incompleteness of the in vitro experiments on cell lines, and those of Reviewer 2 (please see Point #2) about the unpredictability of mirroring in vivo data in cell lines, we have removed Figure 3 from the revised manuscript without compromising the conclusions of this study.

(c) An additional marker for oxidative stress was also raised by Reviewer 1.  In our hands, HNE IF staining provided the most consistent results. Unlike many reagents that can provide neat quantification of oxidative stress in cultured cells, few published works show oxidative stress in FFPE tissue sections. We have done NRF2 IF staining with reasonably good results, but the Ab batch that we used is no longer available and thus could not be reproduced.  A few other NRF2 Ab that we tested did not give satisfactory results. Therefore, we did not show the NRF2 IF data. We had similar issues with GCLC and HO-1 antibodies. At present, we do not have a better indicator of oxidative stress in FFPE tissues and are trying to find a good marker. A potential candidate is 8-hydroxy-2-deoxyguanosine (8-oxo-dG), which optimization for use on liver FFPE sections.

2) The in vitro overexpression experiments of Figure 3 do show that SOCS1 overexpression may suppress CDKN1A, but these results are also slightly puzzling and require more clarification. The authors’ previous work showed that SOCS1 promoted p53 activation (reference 10). It is strange that the p53 target CDKN1A has lower expression when SOCS1 is overexpressed in the liver cell lines. Is this CDKN1A effect dependent on p53 or is this a result of SOCS3 being downregulated when SOCS1 is overexpressed? The authors should show the expression levels of SOCS3 when SOCS1 is overexpressed. Additionally, all the in vivo experiments were done with loss of SOCS1, but the mechanistic in vitro experiments were done with SOCS1 overexpression. We cannot rule out that SOCS1 overexpression and loss may have very different effects. The authors should try to knock out or knock down SOCS1 in the liver cell lines to see if they get a similar downstream effect of CDKN1A and NRF2 activation.

Response: The work presented in Reference 10 (DOI: 10.1016/j.molcel.2009.09.044) was done on cellular senescence in a fibroblast cell line in vitro, which showed that SOCS1 activates p53.

Our work presented in Reference 4 (DOI: 10.1038/onc.2015.485) was done to validate these findings in the livers of SOCS1-deficient mice. This study (Ref 4) showed that SOCS1 is dispensable for p53 activation in the liver. We show now that this is due to a compensatory SOCS3 response in hepatocytes that does not happens in T cells or fibroblasts, which were the cell types studied in the paper (Ref 4) mentioned above. 

We completely agree with the statement of the reviewer: ‘SOCS1 overexpression and loss may have very different effects’. In addition, genetic alterations accumulated in these long-established cell lines may influence the impact of SOCS1 overexpression on Socs3 gene induction and p53 activation. Moreover, Hep3B does not express p53 (Ref: DOI: 10.1007/s10616-014-9761-9). Therefore, trying to answer the above questions in cell lines used in this study may not provide reliable data. The proposed knockout or knockdown SOCS1 approach using other liver cell lines to study the effect of SOCS1/ overexpression on SOCS3, CDKN1A and NRF2 activation is a project on its own that would need several months to complete, and is unlikely to add any new information already gleaned from the in vivo data.

We believe that the already published p21-Nrf2 link (Ref #20: DOI: 10.1016/j.molcel.2009.04.029) and our in vivo data with genetic proofs that clearly demonstrate the ‘SOCS1 loss-p21 induction-NRF2 activation’ pathway (via Socs3 upregulation) are strong enough ‘to proceed from Figure 2 to Figure 4’ without the need for Figure 3. Therefore, we have removed Figure 3 and all the in vitro cell line data. This will eliminate the concerns raised and make the manuscript straight forward without compromising the conclusion of this manuscript. We have modified the manuscript accordingly to make a smooth transition from Figure 2 to Figure 4 (Figure 3 in the revised MS) in the Results section. Appropriate modifications are made in the abstract to reflect these changes.

Minor concern:

1) Figure legend for Figure 4f is missing.

Response: Thank you, we have added this detail in Figure 3 legend in the revised manuscript.

Round 2

Reviewer 2 Report

The authors have addressed all previous concerns. I recommend the article to be published in its present form.